# Radiomics machine learning study with a small sample size: Single random training-test set split may lead to unreliable results

**Chansik An[1,2☯], Yae Won Park[3☯], Sung Soo Ahn[3]\*, Kyunghwa Han[3], Hwiyoung Kim[3], Seung-Koo Lee[3]**

1 Department of Radiology, National Health Insurance Service Ilsan Hospital, Goyang, Korea, 2 Research Institute, National Health Insurance Service Ilsan Hospital, Goyang, Korea, 3 Department of Radiology and Research Institute of Radiological Science and Center for Clinical Imaging Data Science, Yonsei University College of Medicine, Seoul, Korea

☯ These authors contributed equally to this work.
* sungsoo@yuhs.ac

**Data Availability Statement:** All the datasets and codes used in this study can be found on our GitHub repository: https://github.com/Chansikan/do_not_split_small_sample.

## Abstract

This study aims to determine how randomly splitting a dataset into training and test sets affects the estimated performance of a machine learning model and its gap from the test performance under different conditions, using real-world brain tumor radiomics data. We conducted two classification tasks of different difficulty levels with magnetic resonance imaging (MRI) radiomics features: (1) "Simple" task, glioblastomas [n = 109] vs. brain metastasis [n = 58] and (2) "difficult" task, low- [n = 163] vs. high-grade [n = 95] meningio-mas. Additionally, two undersampled datasets were created by randomly sampling 50% from these datasets. We performed random training-test set splitting for each dataset repeatedly to create 1,000 different training-test set pairs. For each dataset pair, the least absolute shrinkage and selection operator model was trained and evaluated using various validation methods in the training set, and tested in the test set, using the area under the curve (AUC) as an evaluation metric. The AUCs in training and testing varied among different training-test set pairs, especially with the undersampled datasets and the difficult task. The mean (±standard deviation) AUC difference between training and testing was 0.039 (±0.032) for the simple task without undersampling and 0.092 (±0.071) for the difficult task with undersampling. In a training-test set pair with the difficult task without undersampling, for example, the AUC was high in training but much lower in testing (0.882 and 0.667, respectively); in another dataset pair with the same task, however, the AUC was low in training but much higher in testing (0.709 and 0.911, respectively). When the AUC discrepancy between training and test, or generalization gap, was large, none of the validation methods helped sufficiently reduce the generalization gap. Our results suggest that machine learning after a single random training-test set split may lead to unreliable results in radiomics studies especially with small sample sizes.

**Funding:** This research received funding from the Basic Science Research Program through the National Research Foundation of Korea (NRF) funded by the Ministry of Science, Information and Communication Technologies & Future Planning (2020R1A2C1003886) by S.S.A. This research was also supported by Basic Science Research Program through the National Research Foundation of Korea (NRF) funded by the Ministry of Education (2020R1I1A1A01071648) by Y.W.P.

**Competing interests:** The authors have declared that no competing interests exist.

## Introduction

Since the advent of precision and personalized medicine, machine learning (ML) has received great interest as a promising tool for identifying the best diagnosis and treatment for an individual patient. ML research has expanded rapidly in various fields, including radiomics—a method to uncover disease characteristics using a large number of features extracted from medical images [1]. According to the PubMed database with the search term "(machine learning OR deep learning) AND radiomics", the number of published papers per year was 2 in 2015 and increased to 556 in 2020 [2].

Millions of observations are often required for an ML model to be robust and reach acceptable performance levels [3]. However, many ML studies with medical image data, including radiomics-based ML, are often conducted in a small group of patients especially when rare diseases are involved, but still reporting promising predictive accuracies [4]. For their potential clinical usefulness to be ascertained, models must be rigorously validated in independent external datasets. However, most published prediction models have not been validated externally [5], and the field of radiomics ML is no exception where the problem is magnified by the intrinsic difficulty of acquiring large datasets. Recent reports showed that external validation was missing in 81–96% of published radiomics-based studies [4, 6, 7].

In the absence of external validation data, one of the most common strategies to prove the validity of a model's performance is randomly splitting an available dataset into training and test sets; a model is first tuned based on estimated performance in the training set and then validated in the test set. However, it has been shown that random data splitting, by reducing sample sizes and adding more randomness, increases the errors in estimated model performance and the instability of validation results, especially with small sample sizes, which may lead to unstable and suboptimal model development, and overly optimistic performance estimation [8–10].

Despite its importance, the stability issue has received little attention in the radiomics ML community, as the main focus has been on model performance. In the field of radiomics-based ML, to the best of our knowledge, no study has investigated the size of the variabilities caused by random data splitting using real-world radiomics data. Specifically, we were interested in exploring how widely the radiomics ML results—model performance estimated in training and that obtained in testing—can vary depending on how a dataset is split into training and test sets, which could be used as caveats when conducting radiomics ML studies and interpreting the results.

Therefore, the purpose of this study was to investigate how splitting a dataset randomly into training and test sets affects the ML model's estimated performance and its difference from internally validated performance using real-world brain tumor radiomics data under different conditions: the number of input features, sample size, and task difficulty.

## Materials and methods

Fig 1 shows the flow of this study. This retrospective study was approved by the Severance Hospital Institutional Review Board, and informed consent from the patients was waived (IRB No. 2020-2996-001). All analyses were performed using Python 3.0 with scikit-learn 0.23.2 and R 4.0.2. The 95% confidence interval (CI) of area under the receiver operating characteristics curve (AUC) in a test set was estimated using the DeLong method, a nonparametric approach by using the theory on *U*-statistics to generate an estimated covariance matrix [11]. Interested readers can also find a detailed explanation on how to implement the DeLong method in a review paper [12]. The difference in value was considered statistically significant when 95%

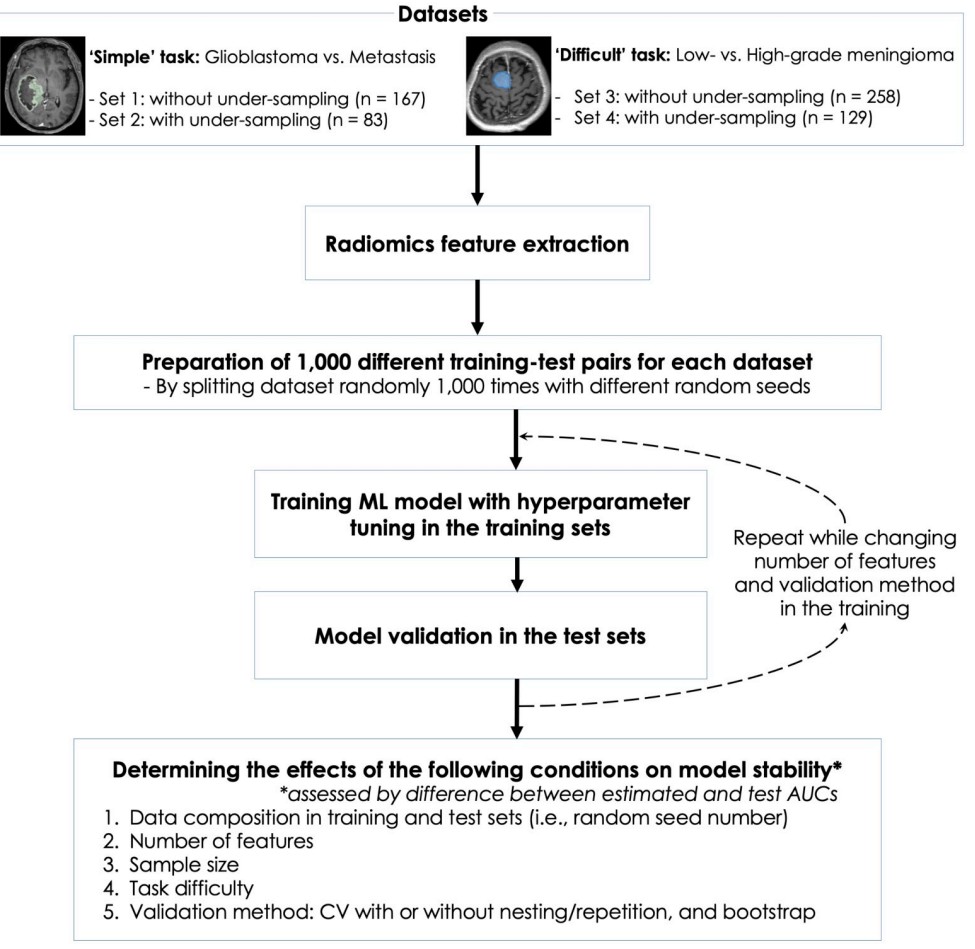

**Fig 1. Study flow.**

CIs did not overlap. All the datasets and codes used in this study can be found on our GitHub repository [13].

## Data

Two classification tasks of different difficulty levels were performed using radiomics features extracted from postcontrast T1-weighted and T2-weighted images of brain magnetic resonance imaging (MRI).

The first task was a "simple" task of differentiating between glioblastoma (GBM) and single metastasis, with the AUC of radiomics-based ML reported to be over 0.9. Chen et al. used postcontrast T1-weighted images and reported an AUC of 0.800 [14]. Bae et al. used postcontrast T1-weighted and T2-weighted images and reported an AUC of 0.956 in the external validation [15]. In the current study, the dataset for the first task consisted of the radiomics features extracted from pathologically confirmed single GBM (n = 109) or single brain metastasis (n = 58) found in 167 adult patients who underwent brain MRI between January 2014 and December 2017.

The second task was a "difficult" task of differentiating between low- and high-grade meningioma by MRI, which is a well-known challenging task to clinicians and radiologists

[16]. Using radiomics features extracted from postcontrast T1-weighted images, Banzato et al. reported an AUC of 0.68 [17], and Chen et al. reported an accuracy of 75.6% [18]. In the current study, the dataset for the second task consisted of the radiomics features extracted from low- (n = 163) or high-grade (n = 95) meningiomas in 258 adult patients who underwent brain MRI between February 2008 and September 2018.

Both the datasets were taken from the same tertiary academic hospital; some subsets of these patients were used in previous studies, where how MRI acquisition, image preprocessing, and radiomics feature extractions were performed for the datasets are described in detail [15, 19]. Additionally, two undersampled datasets were created by randomly sampling 50% of the datasets to determine the effect of sample size. Therefore, our final study subjects consisted of the four datasets: 1) simple task (GBM vs. metastasis) without undersampling, 2) simple task with undersampling, 3) difficult task (low- vs. high-grade meningioma) without undersampling, and 4) difficult task with undersampling.

### Random splitting of data

Each of the four datasets was randomly split into training and test sets with a 7:3 ratio while maintaining the proportions of the outcome classes. This process was repeated while changing the random seed number from 0 to 999 to create 1,000 training-test set pairs with different data compositions.

### Machine learning

Our ML model consisted of three modules: 1) data standardization using the mean and standard deviation derived from the training set, 2) feature selection, and 3) binary classification. For feature selection, based on the results of analysis of variance $F$-tests, the top $k$ radiomics features with highest $F$-statistics (i.e., most relevant features) were selected, where $k$ was a hyperparameter. For classification, we mainly used the least absolute shrinkage and selection operator (LASSO)—one of the least flexible algorithms—to minimize the effect of model selection on the results [20].

### Variability in generalization gap by data splits

We investigated how the gap between the model performance estimated in the training set and the model performance in the test set, which we henceforth call generalization gap, vary among different training-test set pairs [21]. Model performance was represented by AUC in differentiating between GBM and metastasis or between high- and low-grade meningioma. Generalization gap was represented by the difference between the AUC estimated by cross validation (CV) or bootstrap in the training set and the AUC in the test set.

First, we examined the effect of increasing the number of input features on the variability in generalization gap by random data splitting, with other hyperparameters set to the default values. A model was trained in the training set and tested in the test set, followed by estimating the model performance by 5-fold CV in the training set. In each of the 1,000 trials (i.e., training-test set pairs), this process was repeated while increasing $k$ from 1 to 150. The number of features for the lowest generalization gap ranged between 20 and 55 for any of the four datasets, which was used as the grid search range for hyperparameter tuning in the following analyses.

Next, we examined the effects of task difficulty and undersampling on the variability in generalization gap by random data splitting. For each of the 1,000 trials, a model was trained following hyperparameter tuning via a grid search 5-fold CV in the training set and was tested in the test set. The mean value of CV AUCs was used as a model performance estimate. The

hyperparameter grid consisted of 1) number of input variables ranging from 20 to 55 and 2) *C*-value ranging from 0.01 to 10.

## Visual demonstration of data mismatch between training and test sets induced by random data splitting

We attempted to visually demonstrate how a large difference in data composition (i.e., data mismatch) between training and test sets induced by random data splitting can lead to a wide generalization gap. Three trials from the meningioma task were selected as representative cases out of the 1,000 random data splits: two trials with large generalization gaps and one trial with a negligible generalization gap. For each of the three training-test set pairs, a linear support vector machine was fit to the training set and tested in the test set. On a two-dimensional feature space using the top two ($k = 2$) radiomics features, each datapoint was plotted in different colors according to the class. and a decision boundary was drawn, along with the mean CV AUC in the training set and the AUC in the test set.

## Comparison of validation methods

Lastly, we examined whether validation methods other than CV can be helpful in correcting for the inherent optimism in model performance estimates obtained on the same sample used to develop the model. For this purpose, among the trials where CV AUC was higher than test AUC from the dataset for the meningioma grading task, we selected 10 trials with most optimistic estimates (i.e., where the generalization gap was largest; severe data mismatch) and 10 trials with moderately optimistic estimates (i.e., where the generalization gap was in the middle of the range; moderate data mismatch). In these training-test set pairs, we tuned models and estimated their performances using four different methods: 1) 5-fold CV without a repetition, 2) 5-fold CV with 10 repetitions, 3) nested 5-fold CV, and 4) bootstrap with 10 repetitions. For each of the two groups (i.e., moderate vs. severe data mismatch groups), the averages of mean CV AUCs, test AUCs, and their differences were compared among the four validation methods.

In CV with *m* repetitions, CV is repeated *m* times following shuffling data. The nested CV has an inner loop CV nested in an outer CV (S1 Fig). The inner loop is responsible for model selection and hyperparameter tuning, while the outer loop is used for error estimation [22]. The bootstrap is a data resampling method that is known to be good for model selection [23]. Given *n* samples available in the data, *n* samples are randomly chosen with replacement (i.e., the same samples can be chosen). A model is fit to the chosen samples and tested on the remaining (out-of-bag) samples. This process is repeated multiple *B* times (e.g., $B = 10$ in our case), and the out-of-bag scores are averaged as the final estimate of model performance.

## Results

### Variability in generalization gap by data splits

When averaged across the 1,000 different trials (i.e., training-test set pairs), as the number of features increased, both the mean CV AUC (estimated performance) and the test AUC (validated performance) increased in the beginning and later decreased (Fig 2). However, without averaging, in some trials the AUC did not decrease but plateaued with increased number of features, while in some other trials the AUC dropped steeply well below average. Consequently, as the number of features increased beyond a certain point, the variabilities in mean CV AUC and test AUC by random data splits increased. All these variations and trends were more pronounced with the difficult task and with the undersampled datasets (Fig 2).

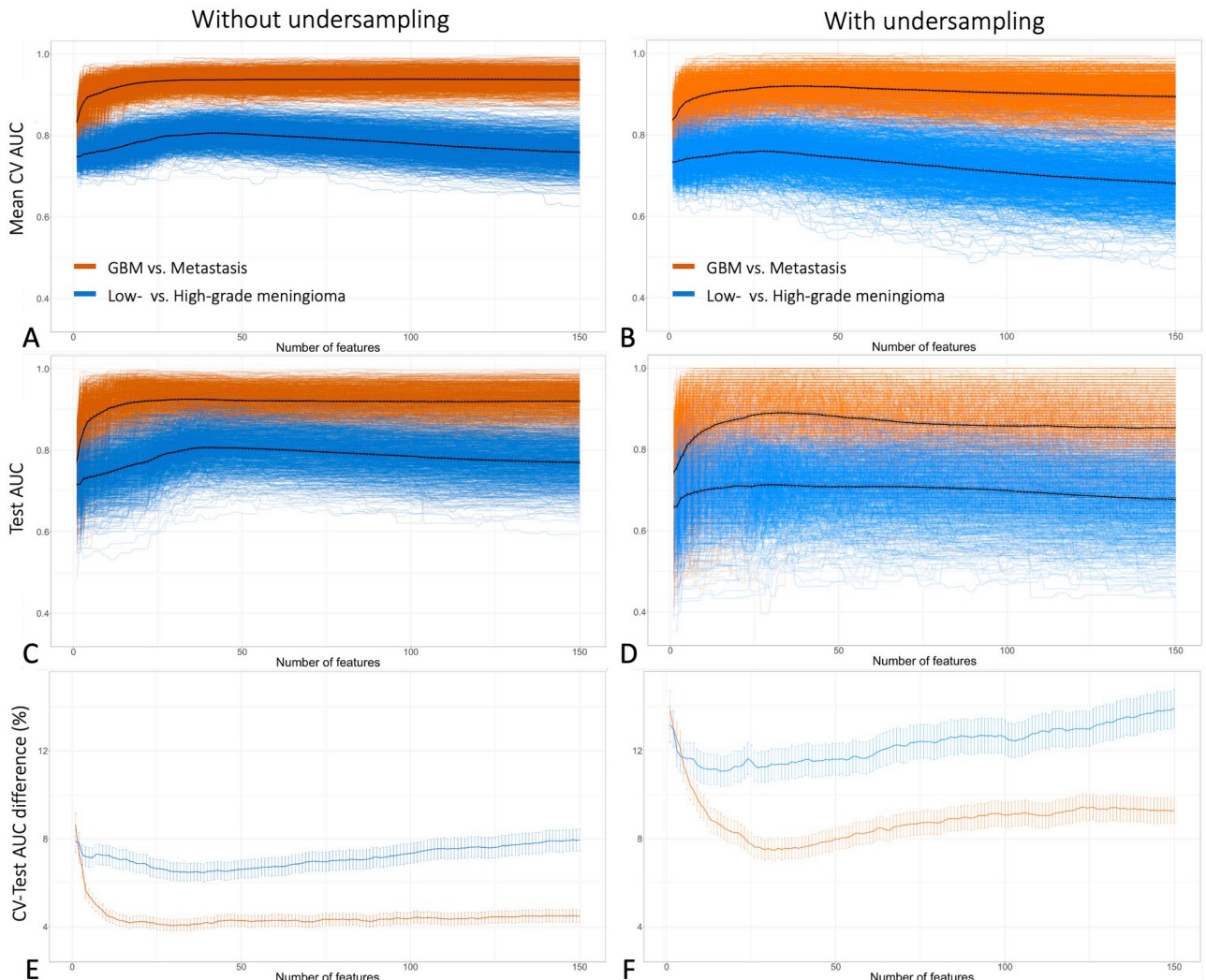

**Fig 2. The relationship between the number of input features and the variability in estimated and test model performance by 1,000 different random data splits. A**, **B, C, and D**: Each colored line indicates a result from a single random training-test set split, and each thick black line shows the average of the results from 1,000 data splits in relation to the number of features: mean cross-validated AUC without (**A**) or with undersampling (**B**), and test AUC without (**C**) or with undersampling (**D**). On average, AUC increased in the beginning and later decreased with increased number of features. However, without averaging, in some trials the AUC did not decrease even at higher numbers of features, widening the AUC variability by the data splits with increased number of features. **E and D**: Each colored line indicates the average of the percentage AUC differences between CV and testing (i.e., generalization gap) from 1,000 data splits in relation to the number of features, with datasets without (**E**) or with undersampling (**F**). The vertical lines indicate 95% confidence intervals. Note that all these trends were more pronounced with the difficult task (i.e., meningioma grading) and with undersampling.

The overall model performance was better, and generalization gap was smaller, with the simple task and without undersampling (Table 1 and Fig 3). The mean test AUC (model performance) and mean AUC difference between CV and testing (generalization gap) were 0.928 and 0.039 for the simple task, and 0.799 and 0.054 for the difficult task, respectively. With undersampling, both the model performance and the generalization gap worsened, with the mean test AUC and the mean AUC difference between CV and testing of 0.879 and 0.079 for the simple task, and 0.697 and 0.092 for the difficult task, respectively.

**Table 1. Model performance estimate and generalization gap according to the sample size and the level of task difficulty.**

| | Simple task (GBM vs. Metastasis) | | Difficult task (Low- vs. High-grade meningioma) | |
|---|---|---|---|---|
| | Without under-sampling | With under-sampling | Without under-sampling | With under-sampling |
| Performance estimation: Mean (±SD) test AUC | 0.928 (±0.038) | 0.879 (±0.072) | 0.799 (±0.045) | 0.697 (±0.073) |
| Generalization gap: Mean (±SD) CV-testing AUC difference | 0.039 (±0.032) | 0.079 (±0.063) | 0.054 (±0.041) | 0.092 (±0.071) |

The values were averages of 1,000 trials with random training-test set splitting. GBM, glioblastoma; SD, standard deviation; AUC, area under the curve; CV, cross validation.

The AUC difference between CV and testing (i.e., generalization gap) varied widely based on samples comprising the training and test sets. Consequently, there were significant discrepancies between expected and actual model performances in some of the trials. Three representative trials for each of the two tasks are summarized in Table 2.

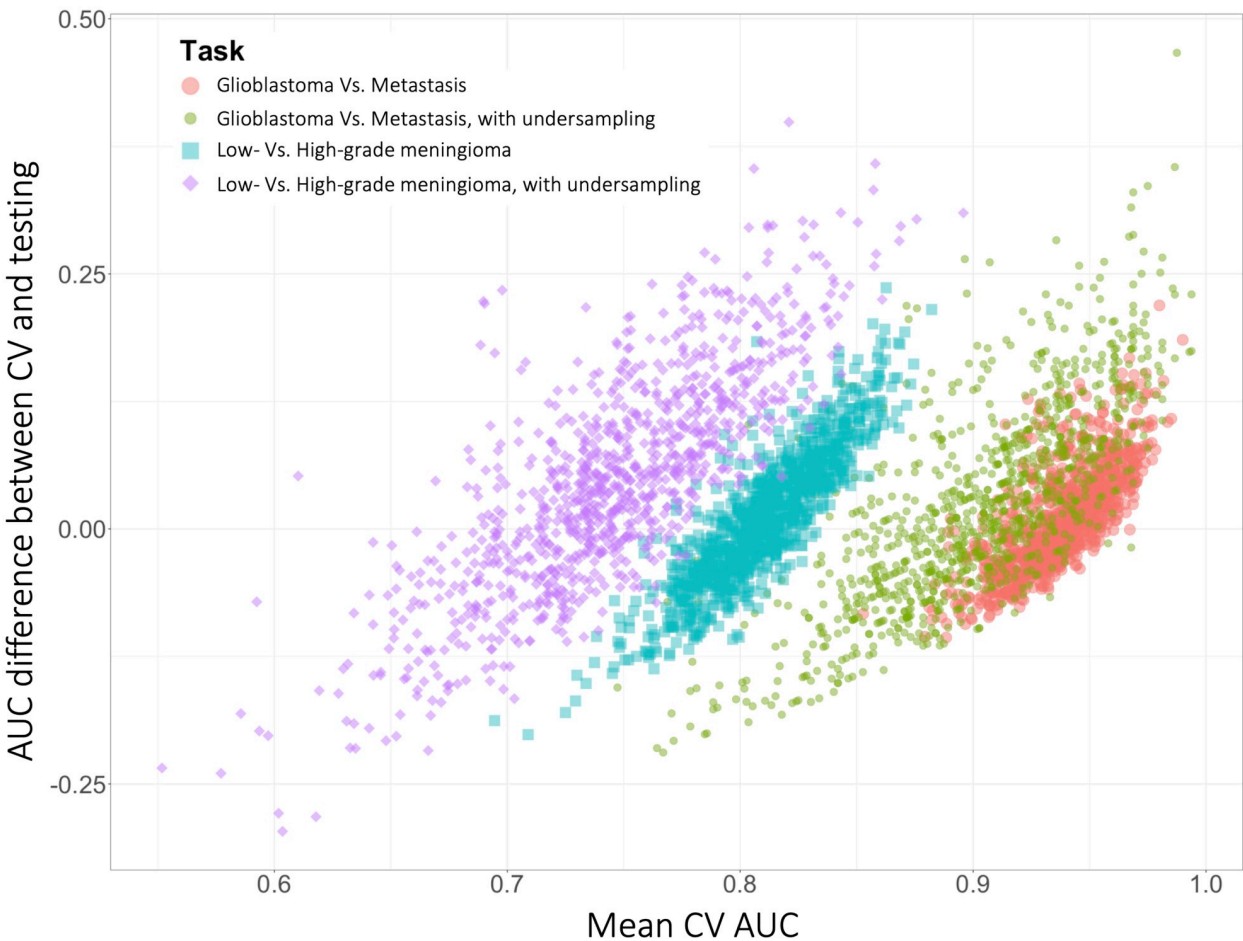

**Fig 3. Model performance estimate and generalization gap in 1,000 different training-test set pairs according to the sample size and the level of task difficulty.** Each point indicates a model performance estimate in the training set (X axis) and its gap from the performance in the test set (Y axis) from a single training-test set pair. Inspection of how closely clustered the datapoints are reveals that the variability in both the model performance estimation and the gap between CV and testing were worse with the difficult task and with undersampling.

**Table 2. Model performance estimate and generalization gap according to the level of task difficulty in some representative training-test set pairs.**

| Training-test sets | Trial No. | Optimal No. of features | Optimal *C*-value | Mean AUC in CV (±SD) | AUC in testing (95% CI) | AUC difference |
|---|---|---|---|---|---|---|
| Simple task: glioblastoma vs. brain metastasis | | | | | | |
| Mismatch, high CV AUC | 995 | 50 | 8 | 0.980 (±0.029) | 0.761 (0.604–0.918) | -0.219 |
| Mismatch, high test AUC | 656 | 40 | 0.09 | 0.888 (±0.083) | 0.995 (0.985–1.000) | +0.107 |
| No mismatch | 733 | 40 | 0.3 | 0.953 (±0.035) | 0.946 (0.890–1.000) | -0.007 |
| Difficult task: low- vs. high-grade in brain meningioma | | | | | | |
| Mismatch, high CV AUC | 346 | 40 | 0.9 | 0.882 (±0.059) | 0.667 (0.539–0.796) | -0.215 |
| Mismatch, high test AUC | 602 | 45 | 0.3 | 0.709 (±0.047) | 0.911 (0.847–0.976) | +0.202 |
| No mismatch | 106 | 45 | 1.5 | 0.830 (±0.046) | 0.821 (0.724–0.919) | -0.009 |

AUC, area under the curve; CV, cross validation; SD, standard deviation; CI, confidence interval.

## Visual demonstration of data mismatch between training and test sets induced by random data splitting

The mechanism behind the large generalization gap is demonstrated by three representative cases chosen among 1,000 trials of differentiating low- and high-grade meningiomas using two radiomic features (Fig 4). In the first case (Trial no. 518), the mean CV AUC was high (0.797 [SD: 0.046]) while the test AUC was low (0.550 [95% CI: 0.414–0.686]). The fitted linear decision boundary separated the two classes well in the training set. However, when applied to the test set, the samples from each class were often located on the wrong sides in the feature space (Fig 4, right panel). In contrast, in the second case (Trial no. 602), the mean CV AUC was low (0.610 [SD:0.042]) while the test AUC was high (0.894 [95% CI: 0.814–0.973]). In the third case (Trial No. 563), the performance discrepancy was negligible (mean CV AUC: 0.711 [SD: 0.046]; test AUC: 0.711 [95% CI: 0.589–0.834]) owing to similar distributions of datapoints in the feature space.

## Comparison of validation methods

In comparison to 5-fold CV without a repetition, 5-fold CV with 10 repetitions, nested 5-fold CV, and bootstrap with 10 repetitions, with increasing order, tended to better correct for the optimism in estimating model performance. However, when there was severe data mismatch between training and test sets, the generalization gap remained large even after the correction (Table 3).

## Discussion

Our results demonstrated that both the radiomics ML model's performance estimated in training and that obtained in testing can vary widely between different training-test set pairs, especially with small sample sizes or difficult tasks. Therefore, the model performance and generalization gap that are estimated after a single random training-test set split may be unreliable when a sample size is not sufficient.

Splitting a dataset into training and test sets has a critical drawback that it reduces the number of samples in both the training and test sets, which leads to suboptimal models and unstable validation results [10, 23, 24]. A previous study demonstrated that a split sample approach with 50% held out led to models with a suboptimal performance, on average similar to the model obtained with half the sample size [23]. Another study showed that small sample sizes

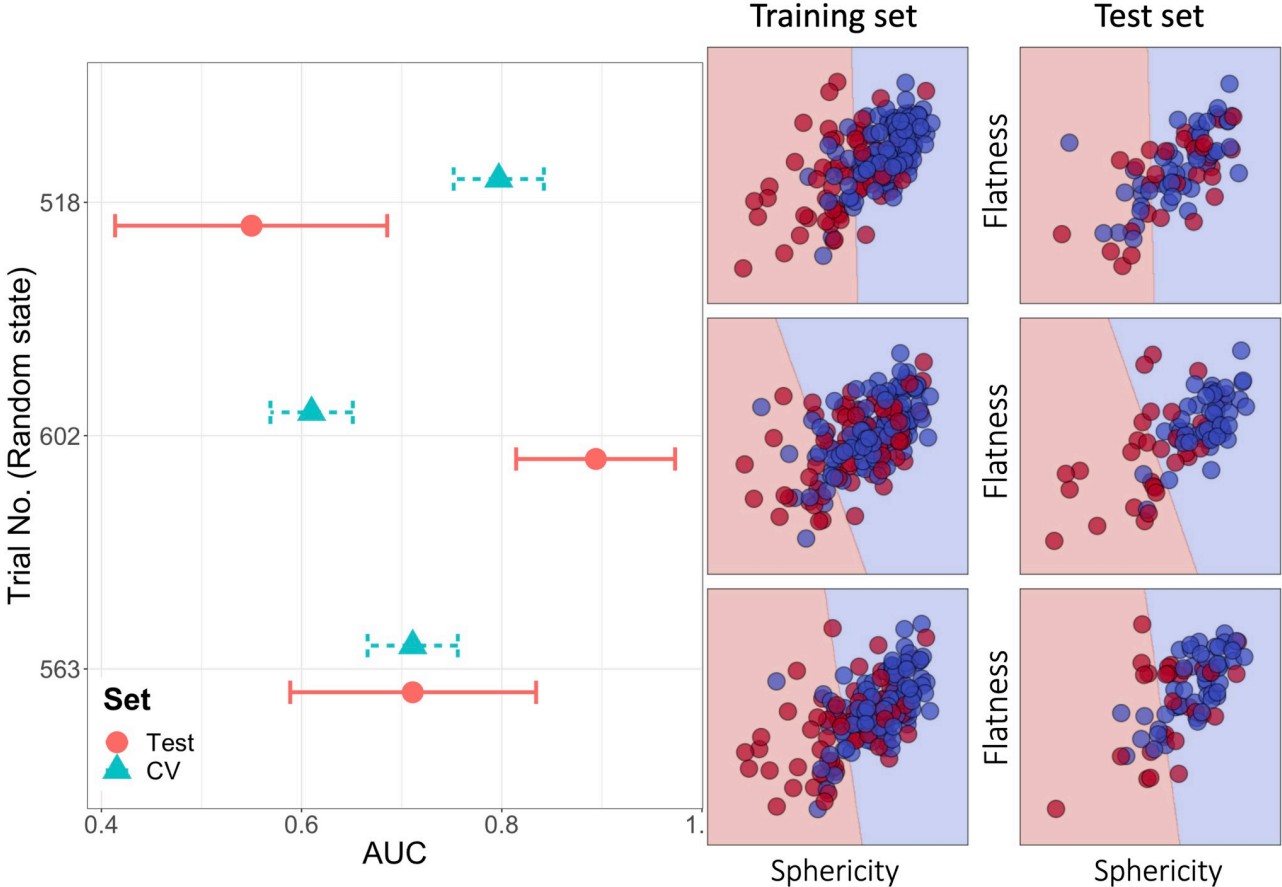

**Fig 4. Discrepancy between performance estimated from cross validation in the training set and performance in the test set (i.e., generalization gap), explained by the distribution of datapoints in the feature space.** The left panel shows that mean cross-validated AUC in the training set and AUC in the test set in three representative trials. The horizontal error bars are standard deviation for CV and 95% confidence interval for testing. The right panel shows the distribution of datapoints in the space consisting of two most important radiomics features (sphericity and flatness). The blue and red dots indicate low- and high-grade meningioma cases, respectively. The linear line dividing the feature space into two areas is a decision boundary; if a datapoint is located in the blue or red area, the model predicts that it is low- or high-grade meningioma, respectively.

**Table 3. Comparison of four methods to estimate model performance in the meningioma grading task.**

|  | Training AUC | Test AUC | AUC difference |
|---|---|---|---|
| In 10 trials with moderate data mismatch | | | |
| CV | 0.824 (0.792, 0.847) | 0.760 (0.731, 0.78) | 0.066 (0.06, 0.074) |
| CV with repetitions | 0.819 (0.775, 0.834) | 0.766 (0.739, 0.811) | 0.051 (-0.023, 0.08) |
| Nested CV | 0.809 (0.741, 0.838) | 0.758 (0.739, 0.78) | 0.045 (0.002, 0.08) |
| Bootstrap | 0.758 (0.741, 0.798) | 0.760 (0.731, 0.78) | 0.005 (-0.021, 0.038) |
| In 10 trials with severe data mismatch | | | |
| CV | 0.861 (0.806, 0.881) | 0.677 (0.629, 0.704) | 0.184 (0.15, 0.232) |
| CV with repetitions | 0.863 (0.783, 0.881) | 0.662 (0.652, 0.685) | 0.187 (0.128, 0.224) |
| Nested CV | 0.844 (0.768, 0.885) | 0.675 (0.62, 0.694) | 0.166 (0.113, 0.226) |
| Bootstrap | 0.806 (0.743, 0.840) | 0.677 (0.629, 0.704) | 0.134 (0.055, 0.182) |

Numbers in cells are median (minimum, maximum). AUC, area under the curve; CV, cross validation.

inherently lead to unreliable CV results with large error bars, meaning that we cannot ascertain whether a cross–validated model will perform in a set of new patients as well as expected [25].

In the current study, despite its larger sample size, the "difficult" meningioma grading task suffered the significantly larger variability in generalization gap by random data splitting than the "simple" GBM-metastasis differentiation task. This observation shows that a "not-too-small" sample size depends on the task difficulty; the harder the ML task is, the more sample we need. This is not a new finding, and it is well known that the sufficient training data size depends on the complexity or difficulty of a task [20]. However, what we attempted to demonstrate is that, ironically, the harder a radiomics ML task is, the more overly optimistic results we can obtain by random data splitting.

By examining the effect of increasing the number of features on the model performance and stability, we intended to demonstrate the necessity to limit the number of input features to a sufficient minimum. When performing a hyperparameter tuning using a large grid of hyper-parameters including the number of features, an unusually high number of features can be selected as optimal simply by chance. However, such model is highly likely to fail to generalize as seen in our result. In general, the higher the ratio of the number of training patterns to the number of classifier parameters, the better the generalization property of the resulting classifier [26]. A larger number of features means a higher number of classifier parameters. Thus, for a limited number of training samples, keeping the number of features as reasonably small as possible is helpful in selecting a classifier with a good generalization capability.

Our results can also be understood by imaging an $n$–dimensional feature space (in this study, two-dimensional spaces were used for the sake of visualization; Fig 4). The Euclidean distance between two random points in the two-dimensional unit square is approximately 0.52 on average. In the three-dimensional unit cube, the average distance is 0.66, and in the ten-dimensional hypercube, it increases to approximately 3.16 [27]. In contrast, the average distance becomes smaller with a larger number of datapoints. Therefore, as the dimension (i.e., the number of features) increases, or the sample size decreases, the datapoints in the feature space are located farther from one another, leading to a greater chance that two randomly split subsets have significantly different data distributions. Additionally, as a classification task becomes more challenging, there are more areas in a feature space where datapoints belonging to different classes are mingled together; the datapoints in these areas are more critical in esti-mating a decision boundary than datapoints in other areas. Thus, in a difficult task with many such datapoints, a decision boundary may vary greatly according to how a dataset is split into training and test sets.

For the above-mentioned reasons and others, some, including us, argue that a random sam-ple split is best avoided especially with a small sample size [10, 23, 28]. Some went a step fur-ther to advocate that external validation should not be included in the model development study and should be performed by different researchers than those involved in the model development [29]. However, it is not realistic to require external validation for all studies, and some argued that many failed external validations could have been foreseen by rigorous inter-nal validation, saving time and resources [30]. Several methods have been proposed to obtain the reliable estimates of true model performance [8, 9, 22, 23, 31], supported by our results that using some of these methods—nested CV or bootstrapping—could reduce the optimism in estimating model performance. However, we also found that a significant data mismatch between training and test datasets can be challenging to overcome by any technique, as others also pointed out [25, 32, 33].

There are limitations in our study. We selected two "simple" and "difficult" ML tasks using brain MRI radiomics features arbitrarily, but they cannot represent all real-world radiomics ML tasks. Another limitation is that we did not perform mathematical analysis in this study.

However, our focus was more on demonstrating the actual consequences of random data splitting on the results of real-world radiomic-based ML studies.

## Conclusions

When a sample size is not sufficient for a radiomics ML task, the model's performance estimated in training and that obtained in testing may vary widely between different training-test set pairs. Therefore, a single random split of a dataset into training and test sets may lead to an unreliable report of the estimated model performance and generalization gap. Furthermore, since the variability of generalization gap tends to be wider with smaller sample sizes and more difficult tasks, ironically, the harder a radiomics ML task is, the more overly optimistic results we can obtain by random data splitting. Therefore, we advise against splitting a small dataset into training and test sets and recommend reducing the optimism in estimating model performance by using bootstrapping, nested CV or other techniques to better predict generalization gap, when external validation is not performed. Future study with real-world data other than brain MRI radiomics is warranted to further investigate the impact of random data splitting on ML study results.

## Supporting information

**S1 Fig. Nested cross validation.** The inner loop is responsible for model selection and hyperparameter tuning, while the outer loop is used for error estimation.
(TIF)

## Author Contributions

**Conceptualization:** Chansik An, Sung Soo Ahn.

**Data curation:** Chansik An.

**Formal analysis:** Chansik An, Kyunghwa Han.

**Investigation:** Chansik An, Yae Won Park, Hwiyoung Kim.

**Methodology:** Chansik An, Sung Soo Ahn, Kyunghwa Han, Hwiyoung Kim.

**Software:** Kyunghwa Han.

**Supervision:** Sung Soo Ahn, Kyunghwa Han, Seung-Koo Lee.

**Validation:** Yae Won Park.

**Visualization:** Chansik An.

**Writing – original draft:** Chansik An, Yae Won Park, Sung Soo Ahn.

**Writing – review & editing:** Chansik An, Yae Won Park, Sung Soo Ahn, Kyunghwa Han, Hwiyoung Kim.

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
