## [Decision Letter · Decision Letter 0]

23 Apr 2021

PONE-D-21-06656

Radiomics machine learning study with a small sample size: single random training-test set split may result in unreliable results

PLOS ONE

Dear Dr. Ahn,

Thank you for submitting your manuscript to PLOS ONE. After careful consideration, we feel that it has merit but does not fully meet PLOS ONE’s publication criteria as it currently stands. Therefore, we invite you to submit a revised version of the manuscript that addresses the points raised during the review process.

We look forward to receiving your revised manuscript.

Kind regards,

Khanh N.Q. Le

Academic Editor

PLOS ONE

Journal Requirements:

"This retrospective study (2020-2996-001 ) was approved by the Institutional Review Board, and the informed consent from the patients was waived."

5. Please upload a copy of Figure S1, to which you refer in your text on page 7. If the figure is no longer to be included as part of the submission please remove all reference to it within the text.

6. Please include a copy of Table S2 which you refer to in your text on page 10.

Reviewers' comments:

Reviewer's Responses to Questions

**Comments to the Author**

1. Is the manuscript technically sound, and do the data support the conclusions?

Reviewer #1: Partly

Reviewer #2: Yes

Reviewer #3: Partly

2. Has the statistical analysis been performed appropriately and rigorously? 

Reviewer #1: No

Reviewer #2: Yes

Reviewer #3: Yes

3. Have the authors made all data underlying the findings in their manuscript fully available?

Reviewer #1: Yes

Reviewer #2: Yes

Reviewer #3: No

4. Is the manuscript presented in an intelligible fashion and written in standard English?

Reviewer #1: Yes

Reviewer #2: Yes

Reviewer #3: Yes

5. Review Comments to the Author

Reviewer #1: 1. The keywords are missing in the abstract.

2. The main contribution of the work must be properly framed in the introduction.

3. The introduction must highlight the actual problem the authors tried to solve.

4. The literature review is very very weak in the article, author needs to append more quality works from 2019-2021.

5. Which datasets the authors have used is not getting clear in the Materials & Methods section, authors need to describe more

about the datasets used.

6. The paper is mathematically weak, authors need to check this issue carefully, they not show the mathematical part also.

7. The proposed work needs to be compared with some existing works from 2019-2021 for understanding the novelty and

efficiency of the proposed work.

8. The discussion needs more elaboration and explanation.

9. Only high definition images must be used in the manuscript. Figure 1,2 needs to be changed with HD images.

Reviewer #2: The manuscript addresses the reliability of radiomics analysis when there are less samples and they are randomly splitted into training and test dataset. The study is of interest and timely.

The manuscript may be accepted for publication.

Reviewer #3: In this paper, the authors tackled an important problem of estimating the classification abilities of machine learning classification models in the case of limited groundtruth data. The topic investigated in this paper is definitely worth considering, but the manuscript suffers from the following shortcomings that need to be addressed before it could be considered for publication:

1. The experiments reported in the paper should be fully reproducible. Hence, the authors should make their MRI data publicly available. If it is not possible due to any reason, the authors should explot a publicly available dataset in their experiments (such as the BraTS dataset). Similarly, I encourage the authors to make their implementation publicly available as well.

2. The authors mentioned the accuracies reported in the literature for the investigated classification problems in the Materials and Methods (Subjects) section — for what data were these accuracies reported?

3. All entities in the paper should be selfcontained — to this end, the authors should expand the captions of their figures and include all the details that will make the figures possible to understand without diving into text. Similarly, please discuss the exploited methods in detail (e.g., De Long method).

4. Although the manuscript reads well in general, I spotted a typo (imagining).

6. PLOS authors have the option to publish the peer review history of their article (what does this mean?). If published, this will include your full peer review and any attached files.

Reviewer #1: **Yes: **Jyotir Moy Chatterjee

Reviewer #2: No

Reviewer #3: No

---

## [Author Response · Author response to Decision Letter 0]

1 Jun 2021

Reviewers’ comments: 

Reviewer #1: 

1. The keywords are missing in the abstract.

>>> We added five keywords (words included in the title were not selected as keywords): prediction model, validation, data splitting, glioblastoma, brain metastasis

2. The main contribution of the work must be properly framed in the introduction.

>>> Thanks to the reviewers’ valuable comments and suggestions, we extensively revised the manuscript (We have practically rewritten the entire manuscript), inspired by the reviewers’ comments and suggestions. In the revised Introduction, the main contribution of our work is summarized as follows:

“Despite its importance, the stability issue has received little attention in the radiomics ML community, as the main focus has been on model performance. In the field of radiomics-based ML, to the best of our knowledge, no study has investigated the size of the variabilities caused by random data splitting using real-world radiomics data. Specifically, we were interested in exploring how widely the radiomics ML results—model performance estimated in training and that obtained in testing—can vary depending on how a dataset is split into training and test sets, which could be used as caveats when conducting radiomics ML studies and interpreting the results.

3. The introduction must highlight the actual problem the authors tried to solve.

>>> In the revised Introduction, the actual problem we tried to solve is highlighted; the following is excerpted from the revised manuscript: 

“… many ML studies with medical image data, including radiomics-based ML, are often conducted in a small group of patients… but still reporting promising predictive accuracies… external validation was missing in 81–96% of published radiomics-based studies… In the absence of external validation data, one of the most common strategies to prove the validity of a model’s performance is randomly splitting an available dataset into training and test sets… random data splitting, by reducing sample sizes and adding more randomness, increases the errors in estimated model performance and the instability of validation results, especially with small sample sizes, which may lead to unstable and suboptimal model development, and overly optimistic performance estimation… the purpose of this study was to investigate how splitting a dataset randomly into training and test sets affects the ML model’s estimated performance and its difference from internally validated performance using real-world brain tumor radiomics.”

4. The literature review is very very weak in the article; author needs to append more quality works from 2019-2021.

>>> Thank you for your comment. We have thoroughly reviewed articles on machine learning and model evaluation methods and updated several references accordingly. Only few relevant methodology papers have been published in recent years, which were still very informative and helped us to improve our manuscript. The following are the papers published within 5 years that we have newly cited in the revised manuscript: 

5. Ramspek CL, Jager KJ, Dekker FW, Zoccali C, Diepen M van. External validation of prognostic models: what, why, how, when and where? Clin Kidney J. 2021;14: 49–58.

6. Won SY, Park YW, Ahn SS, Moon JH, Kim EH, Kang S-G, et al. Quality assessment of meningioma radiomics studies: Bridging the gap between exploratory research and clinical applications. Eur J Radiol. 2021;138: 109673.

7. Won SY, Park YW, Park M, Ahn SS, Kim J, Lee S-K. Quality Reporting of Radiomics Analysis in Mild Cognitive Impairment and Alzheimer’s Disease: A Roadmap for Moving Forward. Korean J Radiol. 2020;21: 1345.

20. Jiang Y, Krishnan D, Mobahi H, Bengio S. Predicting the Generalization Gap in Deep Networks with Margin Distributions. arXiv:1810.00113v2 [Preprint]. 2018 [cited 2021 May 20]. Available from: https://arxiv.org/abs/1810.00113v2

22. Austin PC, Steyerberg EW. Events per variable (EPV) and the relative performance of different strategies for estimating the out-of-sample validity of logistic regression models. Stat Methods Med Res. 2017;26: 796–808. 

24. Varoquaux G. Cross-validation failure: Small sample sizes lead to large error bars. Neuroimage. 2018;180: 68–77.

27. Steyerberg EW, Harrell FE. Prediction models need appropriate internal, internal–external, and external validation. J Clin Epidemiol. 2016;69: 245–247. 

28. Siontis GCM, Ioannidis JPA. Response to letter by Forike et al.: more rigorous, not less, external validation is needed. J Clin Epidemiol. 2016;69: 250–251. 

29. Martens FK, Kers JG, Janssens ACJW. External validation is only needed when prediction models are worth it (Letter commenting on: J Clin Epidemiol. 2015;68:25-34). J Clin Epidemiol. 2016;69: 249–250.

32. Ho SY, Phua K, Wong L, Goh WWB. Extensions of the External Validation for Checking Learned Model Interpretability and Generalizability. Patterns. 2020;1: 100129.

33. Olsen MK, Stechuchak KM, Steinhauser KE. Comparing internal and external validation in the discovery of qualitative treatment-subgroup effects using two small clinical trials. Contemp Clin Trials Commun. 2019;15: 100372.

5. Which datasets the authors have used is not getting clear in the Materials & Methods section, authors need to describe more about the datasets used.

>>> We have added more description of the datasets used. In addition, we have uploaded all the radiomics datasets and R/Python codes used for this study on our GitHub repository so that anyone can reproduce our results.

URL: https://github.com/Chansikan/do_not_split_small_sample

6. The paper is mathematically weak, authors need to check this issue carefully, they not show the mathematical part also.

>>> Thank you for your valuable comment. We also discussed with a biostatistician. However, we decided that delving into mathematical aspects would be beyond our capabilities (we were afraid that we would end up conveying inaccurate information) and that it would be better to focus on demonstrating the actual consequences with real-world radiomic-based ML studies. Instead, we explicitly explained this limitation in Discussion.

7. The proposed work needs to be compared with some existing works from 2019-2021 for understanding the novelty and efficiency of the proposed work.

>>> In the revised Introduction and Discussion sections, we explained what distinguishes our study from others, although only few papers have been published in recent years, as mentioned above. For examples, the following excerpted from the revised manuscript:

“In the field of radiomics-based ML, to the best of our knowledge, no study has investigated the size of the variabilities caused by random data splitting using real-world radiomics data.”

“… it is well known that the sufficient training data size depends on the complexity or difficulty of a task... However, what we attempted to demonstrate is that, ironically, the harder a radiomics ML task is, the more overly optimistic results we can obtain by random data splitting.”

“Several methods have been proposed to obtain the reliable estimates of true model performance, supported by our results that using some of these methods—nested CV or bootstrapping—could reduce the optimism in estimating model performance. However, we also found that a significant data mismatch between training and test datasets can be challenging to overcome by any technique…”

8. The discussion needs more elaboration and explanation.

>>> As mentioned above, we rewrote nearly the whole manuscript, which we believe improved the manuscript much. The revised Discussion now includes detailed explanation and interpretation of our results.

9. Only high definition images must be used in the manuscript. Figure 1,2 needs to be changed with HD images.

>>> We have changed the Figures 1 and 2 to high-resolution images and uploaded them.

Reviewer #2: The manuscript addresses the reliability of radiomics analysis when there are less samples and they are randomly splitted into training and test dataset. The study is of interest and timely.

The manuscript may be accepted for publication.

>>> We would like to express our gratitude to your positive review. 

Reviewer #3: In this paper, the authors tackled an important problem of estimating the classification abilities of machine learning classification models in the case of limited ground truth data. The topic investigated in this paper is definitely worth considering, but the manuscript suffers from the following shortcomings that need to be addressed before it could be considered for publication:

1. The experiments reported in the paper should be fully reproducible. Hence, the authors should make their MRI data publicly available. If it is not possible due to any reason, the authors should explot a publicly available dataset in their experiments (such as the BraTS dataset). Similarly, I encourage the authors to make their implementation publicly available as well.

>>> We uploaded all the radiomics datasets and R/Python codes used for this study on our GitHub repository (also mentioned in the revised manuscript), so that anyone can fully reproduce our results.

URL: https://github.com/Chansikan/do_not_split_small_sample

2. The authors mentioned the accuracies reported in the literature for the investigated classification problems in the Materials and Methods (Subjects) section — for what data were these accuracies reported?

>>> Thank you for this valuable question. In the revised manuscript, we explained in more detail the characteristics of the datasets from which the accuracies were obtained, by describing the MRI sequences used, and citing a reference separately for each reported accuracy so that readers can refer to it if they’re interested in more information.

3. All entities in the paper should be self-contained — to this end, the authors should expand the captions of their figures and include all the details that will make the figures possible to understand without diving into text. Similarly, please discuss the exploited methods in detail (e.g., De Long method). 

>>> Thank you for pointing this out. We added detailed explanation in the figure legends so that the figures can be understood without diving into the text. 

>>> Also, we revised the manuscript so that readers can understand the methods (i.e., CV with repetitions, nested CV, bootstrap, De Long method) used in our study by reading the text or referring to the references. Furthermore, since now we have made our codes publicly available, anyone can actually implement those methods by themselves.

4. Although the manuscript reads well in general, I spotted a typo (imagining). 

>>> We have corrected the typo and some other awkward phrases.

---

## [Decision Letter · Decision Letter 1]

11 Jul 2021

PONE-D-21-06656R1

Radiomics machine learning study with a small sample size: single random training-test set split may lead to unreliable results

PLOS ONE

Dear Dr. Ahn,

Thank you for submitting your manuscript to PLOS ONE. After careful consideration, we feel that it has merit but does not fully meet PLOS ONE’s publication criteria as it currently stands. Therefore, we invite you to submit a revised version of the manuscript that addresses the points raised during the review process.

We look forward to receiving your revised manuscript.

Kind regards,

Khanh N.Q. Le

Academic Editor

PLOS ONE

Journal Requirements:

Reviewers' comments:

Reviewer's Responses to Questions

**Comments to the Author**

1. If the authors have adequately addressed your comments raised in a previous round of review and you feel that this manuscript is now acceptable for publication, you may indicate that here to bypass the “Comments to the Author” section, enter your conflict of interest statement in the “Confidential to Editor” section, and submit your "Accept" recommendation.

Reviewer #1: All comments have been addressed

Reviewer #3: All comments have been addressed

2. Is the manuscript technically sound, and do the data support the conclusions?

Reviewer #1: Yes

Reviewer #3: Yes

3. Has the statistical analysis been performed appropriately and rigorously? 

Reviewer #1: N/A

Reviewer #3: Yes

4. Have the authors made all data underlying the findings in their manuscript fully available?

Reviewer #1: No

Reviewer #3: Yes

5. Is the manuscript presented in an intelligible fashion and written in standard English?

Reviewer #1: Yes

Reviewer #3: Yes

6. Review Comments to the Author

Reviewer #1: Although the manuscript is revised by the authors as per previous comments, but still certain points needs to be considered for modification before considering the paper which is as follows:

1. The dataset citation is still missing in the paper.

2. The authors are suggested to make a tabular format for comparison of the proposed with existing works.

3. The conclusion is very short, authors need to revise it highlighting the research result and the possible future work.

Reviewer #3: I am happy to see that the authors have addressed my concerns. I would, however, still encourage the authors to present the De Long method in more detail in the main body of the manuscript.

7. PLOS authors have the option to publish the peer review history of their article (what does this mean?). If published, this will include your full peer review and any attached files.

Reviewer #1: **Yes: **Jyotir Moy Chatterjee

Reviewer #3: No

---

## [Author Response · Author response to Decision Letter 1]

20 Jul 2021

Reviewer #1: Although the manuscript is revised by the authors as per previous comments, but still certain points needs to be considered for modification before considering the paper which is as follows:

1. The dataset citation is still missing in the paper.

>>> Thank you for your comment. All the raw datasets and (R and Python) codes can be found in a publicly open repository which we have cited in our paper as follows:

“All the datasets and codes used in this study can be found on our GitHub repository [13].” (Page 5 Line 104)

13. An C. GutHub page: do not split small samples; 2021 [cited 30 May 2021]. Available from: https://github.com/Chansikan/do_not_split_small_sample)

2. The authors are suggested to make a tabular format for comparison of the proposed with existing works.

>>> Thank you for your valuable comment. We have thoroughly searched in PubMed MEDLINE and EMBASE databases to identify papers with similar purposes to ours. We believe we tried our best. However, unfortunately, we could not find a study that had the similar objective as ours (to prove that random splitting of training and test sets may lead to unreliable results) in the radiomics machine learning filed. In fields other than radiomics, we could not find a paper with the same purpose either, although there were papers where it was only secondary aim or side results, which we already cited in the manuscript [10, 23, 24, 25, 28]. If the reviewer suggests which papers can be compared to the current work, we would be happy to review them and make a tabular format for comparison. Again, we thank you for your comment to improve the overall quality of our manuscript.

3. The conclusion is very short, authors need to revise it highlighting the research result and the possible future work.

>>> As you have advised, we have extended the conclusion and highlighted the research result, and mentioned possible future work in the revised manuscript as follows:

“When a sample size is not sufficient for a radiomics ML task, the model’s performance estimated in training and that obtained in testing may vary widely between different training-test set pairs. Therefore, a single random split of a dataset into training and test sets may lead to an unreliable report of the estimated model performance and generalization gap. Furthermore, since the variability of generalization gap tends to be wider with smaller sample sizes and more difficult tasks, ironically, the harder a radiomics ML task is, the more overly optimistic results we can obtain by random data splitting. Therefore, we advise against splitting a small dataset into training and test sets and recommend reducing the optimism in estimating model performance by using bootstrapping, nested CV or other techniques to better predict generalization gap, when external validation is not performed. Future study with real-world data other than brain MRI radiomics is warranted to further investigate the impact of random data splitting on ML study results.”

Reviewer #3: I am happy to see that the authors have addressed my concerns. I would, however, still encourage the authors to present the De Long method in more detail in the main body of the manuscript.

>>> Thank you for your comment. As suggested, first we tried to summarize how the method works in several sentences, but it would take at least a couple of paragraphs, which we don’t think is appropriate. Then we tried to explain the method in a separate supplementary material. However, it would end up just copying (with some modification) the contents from a review paper. Therefore, we decided to cite an excellent, easy-to-follow review paper that explains in detail the De Long method and demonstrates how it is calculated. In the main body the manuscript, we added the following sentence with citing the paper: “Interested readers can also find a detailed explanation on how to implement the DeLong method in a review paper [12].”

[12] Hanley JA, Hajian-Tilaki KO. Sampling variability of nonparametric estimates of the areas under receiver operating characteristic curves: An update. Acad Radiol. 1997;4(1):49-58. doi:10.1016/s1076-6332(97)80161-4

---

## [Editor Report · Decision Letter 2]

2 Aug 2021

Radiomics machine learning study with a small sample size: single random training-test set split may lead to unreliable results

PONE-D-21-06656R2

Dear Dr. Ahn,

We’re pleased to inform you that your manuscript has been judged scientifically suitable for publication and will be formally accepted for publication once it meets all outstanding technical requirements.

Kind regards,

Khanh N.Q. Le

Academic Editor

PLOS ONE
---

## [Editor Report · Acceptance letter]

4 Aug 2021

PONE-D-21-06656R2 

Radiomics machine learning study with a small sample size: Single random training-test set split may lead to unreliable results 

Dear Dr. Ahn:

I'm pleased to inform you that your manuscript has been deemed suitable for publication in PLOS ONE. Congratulations! Your manuscript is now with our production department. 

Kind regards, 

on behalf of

Dr. Khanh N.Q. Le 

Academic Editor

PLOS ONE